# Therapeutic Potential of Chlorogenic Acid in Chemoresistance and Chemoprotection in Cancer Treatment

**DOI:** 10.3390/ijms25105189

**Published:** 2024-05-10

**Authors:** Nicole Cortez, Cecilia Villegas, Viviana Burgos, Leandro Ortiz, Jaime R. Cabrera-Pardo, Cristian Paz

**Affiliations:** 1Laboratory of Natural Products & Drug Discovery, Center CEBIM, Department of Basic Sciences, Faculty of Medicine, Universidad de La Frontera, Temuco 4780000, Chile; n.cortezsalvo01@gmail.com (N.C.); c.villegas04@ufromail.cl (C.V.); 2Departamento de Ciencias Biológicas y Químicas, Facultad de Recursos Naturales, Universidad Católica de Temuco, Rudecindo Ortega, Temuco 4780000, Chile; vburgos@uct.cl; 3Instituto de Ciencias Químicas, Facultad de Ciencias, Universidad Austral de Chile, Valdivia 5110566, Chile; leandro.ortiz@uach.cl; 4Laboratorio de Química Aplicada y Sustentable, Departamento de Química, Facultad de Ciencias, Universidad de Tarapacá, Arica 1000000, Chile; jacabrera777@gmail.com

**Keywords:** chlorogenic acid, CGA, chemoresistance, chemotherapy-induced toxicity

## Abstract

Chemotherapeutic drugs are indispensable in cancer treatment, but their effectiveness is often lessened because of non-selective toxicity to healthy tissues, which triggers inflammatory pathways that are harmful to vital organs. In addition, tumors’ resistance to drugs causes failures in treatment. Chlorogenic acid (5-caffeoylquinic acid, CGA), found in plants and vegetables, is promising in anticancer mechanisms. In vitro and animal studies have indicated that CGA can overcome resistance to conventional chemotherapeutics and alleviate chemotherapy-induced toxicity by scavenging free radicals effectively. This review is a summary of current information about CGA, including its natural sources, biosynthesis, metabolism, toxicology, role in combatting chemoresistance, and protective effects against chemotherapy-induced toxicity. It also emphasizes the potential of CGA as a pharmacological adjuvant in cancer treatment with drugs such as 5-fluorouracil, cisplatin, oxaliplatin, doxorubicin, regorafenib, and radiotherapy. By analyzing more than 140 papers from PubMed, Google Scholar, and SciFinder, we hope to find the therapeutic potential of CGA in improving cancer therapy.

## 1. Introduction

Caffeic acid, a member of the phenylpropanoid family, belongs to a diverse group of compounds derived from the carbon skeleton of phenylalanine. These compounds play a crucial role in plant defense, structural support, and overall survival. Within this family, chlorogenic acid (CGA, 5-caffeoylquinic acid, Figure 1) stands out as a noteworthy natural caffeic acid ester derivative. In this review, our focus will be on CGA, a polyphenol with a wide distribution throughout the plant kingdom, particularly in functional foods such as propolis, coffee, Mate tea, and medicinal herbs such as balm, self-heal, rosemary, and sage. Notably, CGA is orally bioavailable, showcasing various activities in the prevention and treatment of several pathologies, including certain types of cancer [1].

Cancer represents a significant global health challenge, contributing to one in six deaths worldwide. The management of cancer is multifaceted, with conventional approaches typically involving surgical tumor resection followed by adjuvant X-ray radiotherapy and/or chemotherapy. Among these interventions, surgery stands out as particularly effective, especially in the early stages of disease progression. Chemotherapy, meanwhile, remains a cornerstone of cancer treatment, and is widely utilized due to its efficacy. However, a major obstacle in chemotherapy is the development of drug resistance, in which cancer cells that are initially susceptible to anticancer agents become resistant over time. This resistance primarily arises from reduced drug uptake and increased drug efflux mechanisms within the cancer cells. Moreover, while chemotherapy has significantly improved patient outcomes, its broad cytotoxic effects inevitably lead to damage to healthy, rapidly dividing cells. Similar concerns are observed with radiation therapy, which can cause harm to adjacent healthy tissues and organs [2].

The anti-inflammatory, antioxidant, and anticancer properties of CGA make it a valuable component in the quest for better health [3]. These attributes have mostly been associated with cancer prevention [4,5]. However, recent studies show that CGA may not only prevent cancer but may also be used to combat chemoresistance; this is a critical aspect given that treatment failure is responsible for up to 90% of cancer-related deaths [6]. This compound has also been reported to be a chemoprotectant against toxicity induced by different drugs used in chemotherapy treatments.

Chemoresistance is a complex phenomenon with multiple interconnected mechanisms that contribute to disease relapse and metastasis [7,8]. Cisplatin (CP) is a vital chemotherapeutic agent in the clinical setting, and is employed for treating various solid tumors such as those affecting the head and neck, testis, ovary, cervix, bladder, and stomach, as well as microcytic and non-small cell lung cancers [9]. In ovarian cancer, the combination of carboplatin (CB) or CP with paclitaxel (PXT) stands as the standard regimen; however, the high relapse rate observed in approximately 50% of patients [10] is attributed to chemotherapy resistance. Gastric cancer faces a similar challenge, with over 50% of operated patients experiencing local relapse, and the 5-year survival rate dropping below 10% in the metastatic setting [11]. Notably, almost all patients with advanced gastric cancer eventually develop resistance to Fluorouracil (5-FU)—the first-line drug in this type of cancer, alongside CP and epirubicin (EPI)—according to the National Comprehensive Cancer Network^®^ (NCCN) guidelines [12]. Chemotherapy poses additional challenges, including non-selective toxicity, which often lead to damage in healthy tissues [13]. This significantly diminishes patients’ quality of life, restricts treatment tolerability, and may result in permanent effects [14]. Among the serious dose-limiting side effects of CP chemotherapy, nephrotoxicity and severe renal dysfunction stand out, alongside unwanted effects like ototoxicity, gastrointestinal toxicity, myelosuppression, ovarian toxicity, and neurotoxicity [15].

CGA emerges as a promising solution, as it increases or suppresses the expression of various markers, including the increase of tumor protein p53 (p53). This protein acts as a tumor suppressor, which means that it regulates cell division by keeping cells from growing and dividing (proliferating) too fast or in an uncontrolled way [16]. It decreases active caspase 3 (CASP3) and Bcl-2-associated X protein (BAX) in normal tissue. CASP 3 is a small protein that is composed of two subunits—a 12- and 17-kDa subunit—and it plays a critical role in programmed cell death. BAX is a protein of the Bcl-2 family and a core regulator of the intrinsic pathway of apoptosis. CGA also affects cyclin D1, and microtubule-associated protein 1 light chain 3 isoform B (LC3B); effectively reducing apoptosis and autophagy in renal tissue. Cyclin D1 acts as a regulatory subunit of a complex formed with CDK4 or CDK6; therefore, its activity is necessary for the G1/S transition of the cell cycle, and LC3B is a central protein in the autophagy pathway. Furthermore, CGA reduces inflammation levels in renal tissues by down-regulating tumor necrosis factor-alpha (TNF-α) and cyclooxygenase-2 (COX-2), with its anti-inflammatory effect associated with the down-regulation of nuclear factor kappa-light-chain-enhancer of activated B cells (NF-κB) [17]. NF-kB is a nuclear transcription factor that regulates the expression of several genes, including genes involved in the immune and inflammatory responses. TNF-α is a key proinflammatory cytokine that triggers inflammatory responses in the body. TNF-α has been shown to activate the NF-kB, resulting in increased transcription of inflammatory genes, including COX-2. COX-2 is an enzyme that catalyzes the conversion of fatty acids into prostaglandins, which are important mediators of inflammation [18].

On the other hand, 5-FU has been linked to significant toxicity, marked by increased oxidative stress and inflammation [19,20]. Its side effects encompass nephrotoxicity, cardiotoxicity, myelosuppression, and ovarian tissue toxicity [21]. CGA plays a pivotal role in mitigating 5-FU-induced nephrotoxicity by replenishing the antioxidant arsenal, effectively down-regulating apoptotic and inflammatory tissue damage caused by 5-FU. Moreover, CGA exhibits a protective effect against 5-FU-induced ovarian tissue damage, reducing Interleukin 6 (IL-6) levels; another cytokine associated with proinflammatory processes linked to ovarian tissue damage caused by 5-FU. While several chemotherapeutic drugs present side effects with distinct molecular mechanisms, often targeting different organs, this discussion will focus on collecting all the evidence for the chemoprotective role played by CGA in specific cases [20,22].

Moreover, CGA suppresses the expression of Programmed Cell Death Ligand 1 (PD-L1) on cancer cells, boosting the antitumor effect of the anti-PD-1 antibody and enhancing anticancer immunotherapy [23].

Numerous studies have investigated the diverse pharmacological properties of chlorogenic acid (CGA), highlighting its antioxidant, hepatoprotective, cardioprotective, and neuroprotective effects. Moreover, CGA exhibits anti-inflammatory, antimicrobial, antiviral, and antihypertensive activities. Its broad spectrum of activity against various microorganisms—encompassing bacteria, yeasts, molds, viruses, and amoebae—presents potential applications in the food industry for the preservation of food products. Additionally, the antioxidant attributes of CGA hold promise for food preservation, particularly in inhibiting lipid oxidation. Consequently, CGA may emerge as a novel and valuable nutraceutical in the foreseeable future [24,25].

This review aims to provide evidence for the biological functions and therapeutic applications of CGA as a chemoprotectant and as a chemosensitizer against cancer chemoresistance. It explores the mechanisms of action of CGA, elucidating its potential clinical efficacy as an adjuvant to chemotherapy to combat resistance, and its crucial role as a chemoprotectant to safeguard healthy tissues against chemotherapy-induced toxicity. Additionally, it offers a concise summary of phytochemical aspects, encompassing functional nutrition, biosynthesis, absorption, distribution, and metabolism of this compound.

## 2. Methodology

A systematic literature review was conducted using prominent databases—including Google Scholar, PubMed, Scopus, and Springer—to compile existing information on the chemosensitizing and chemoprotective roles of CGA. The review focused on in vitro and in vivo analyses, excluding articles primarily discussing chemoprevention. Key search terms included “chlorogenic acid” along with terms such as chemoresistance, chemoprotection, antioxidant effects, toxicity, biosynthesis, metabolism, and food. Specific information about chemotherapy and various toxicities, such as nephrotoxicity, was also included in the search criteria for each section.

## 3. Functional Foods High in Chlorogenic Acid

CGA, also called 5-*O*-caffeoylquinic acid (5-CQA, Figure 1), is a type of polyphenol that is widely distributed in the plant kingdom. Different foods have shown high concentrations of this compound, conferring important biological activities to these functional foods. For example, Table 1 shows the CGA content of some vegetables.

The following vegetables showed no presence of CGA (detection limit = 0.005 mg/kg): fresh pea, green bean, Lablab bean, beet, cassava, turnip, radish, ginger, purple yam, asparagus, celery, green bell pepper, cucumber, bitter melon, purple garlic, and arugula [30].

Coffee, a widely consumed stimulant beverage, owes its stimulating properties to a high caffeine concentration (1000 to 1800 mg/100 g). Cultivated in over 60 tropical countries, coffee beans are rich in various compounds such as alkaloids, polyphenols, and terpenoids [34,35]. Among these, CGA stands out as a crucial polyphenol. Its levels are more pronounced in green coffee (4661 to 4946 mg/100 g) compared to darker roasts (234 to 377 mg/100 g) [26]. In infusions of *C. arabica*, CGA ranged from 628 to 1040 mg/L, similar to *C. canephora* infusions where its concentration was determined to be between 682 and 1210 mg/L. Decaffeinated Arabica green coffee infusion showed a concentration of 767 mg/L [36]. CGA in green coffee beans from (*Coffea arabica*) Brazil, Rwanda, China, Laos; and *Coffea robusta* from Vietnam, India Indonesia, Laos, and Uganda, was extracted for analysis by infusion in distilled water (94 °C/10 min). The highest concentration of CGA was observed in *robusta* type coffee (Vietnam decaffeinated 221.4 g/kg), while the *arabica* type from Rwanda showed the highest concentration (160.1 g/kg); moreover, it was evidenced that the extraction process by steaming the coffee beans decreased the CGA content, but the decaffeination process did not affect it [37].

In the beverage preparation of *Coffea arabica* from South America, both filtration and moka methods were evaluated for their chlorogenic acid content. The results indicated that the highest chlorogenic acid content ranged from 2.94% *w/w* with the filtration method to 3.36% *w/w* with the moka method. These findings suggest that the extraction of CGA is influenced by factors such as duration and maceration during the coffee beverage preparation process [38].

Coffee consumption has been linked to significant bioactivities; for example, coffee polyphenols, particularly CGA, have shown promise in preventing cognitive dysfunction and suppressing amyloid β plaques [39]. Additionally, coffee has demonstrated potential anti-diabetic effects [40] and has been associated with protective effects against certain cardiovascular risk factors [41]. Moreover, coffee is linked to a potential reduction in the risk of various cancers, including breast, colorectal, colon, endometrial, and prostate cancers [42].

Mate tea (*Ilex paraguariensis*)—a beverage with a long history of consumption in South America, particularly in Brazil, Argentina, Paraguay, and Uruguay—has been valued for centuries. Mate tea exhibits notable health benefits, and it is rich in antioxidants and polyphenols, including CGA. The optimal parameters for the preparation of the Mate tea infusion to obtain the highest CGA extraction were determined as 2 g of Mate tea in 300 mL of water (95 °C) by infusion for 16 min [43].

The polyphenol content in sun-exposed Mate tea plants from plantations surpasses that of shaded Mate from forests, with higher levels observed in all polyphenols [27]. It is estimated that a cup of Mate tea, containing one teaspoon (5 g) of instant Mate tea, provides an intake of approximately 1.5 g of gallic acid [44]. The potential antioxidant properties of Mate tea can be further enhanced by enzymatic biotransformation catalyzed by *Paecilomyces variotii tannase*, leading to a 43% increase [45]. In vitro studies demonstrated that Mate tea inhibits colon cancer cell proliferation in human colorectal adenocarcinoma cells CaCo-2 and HT-29, with growth inhibition at Median Growth Inhibition (GI_50_) values of 1.0 and 105.2 µg/mL, respectively [44]. In a 60-day intervention study involving individuals with type 2 diabetes mellitus and pre-diabetic volunteers who consumed 1 L/day of Mate tea, enhanced antioxidant defense and reduced lipid peroxidation activity were observed, indicating mitigation of oxidative stress in all participants [46]. Additionally, hyperlipidemic patients who ingested 200 mL of Mate tea (12.5 mg/mL) daily experienced a significant increase in serum total antioxidant status and the enzymatic activity of superoxide dismutase (SOD), demonstrating an antioxidant-defensive response to low-density lipoprotein (LDL) oxidation [47].

Carrots (*Daucus carota*) are widely consumed root vegetables globally, which are prized for their versatility and nutritional benefits. They boast high levels of carotenoids and phenolic compounds such as CGA, making them valuable ingredients for various applications. For instance, they could be utilized in the development of biofortified beverages aimed at combating obesity, providing antioxidant protection, and reducing inflammation [48]. Moreover, different postharvest treatments have been shown to increase the content of bioactive compounds such as CGA; i.e., wounding stress promotes the formation of CGA and its isomers 3-*O*-caffeoylquinic acid, and 4-*O*-caffeoylquinic acid, as well as the dichloroquinic derivatives 3,5-*O*-dicaffeoylquinic acid, and 4,5-*O*-dicaffeoylquinic acid [49]. In addition, high hydrostatic pressure induces the release of ATP from cells and enzyme activation due to cell-wall disruption caused by pressurization [50], promoting the production of phenolic compounds. For example, whole carrots treated at 100 MPa showed accumulation of 3,4-O-di feruloyl quinic acid (466.1%) and chlorogenic acid (291.2%) after 24 h of storage [51]. Thus, carrots are a versatile ingredient for creating healthy food [52].

Despite the content of CGA in food or beverages, the thermal stability in water of this compound is controversial, due to its isomerization to 3-*O*-caffeoylquinic acid and 4-*O*-caffeoylquinic acid, and further transformation by hydrolysis and water addition to the double bond [53]. For that reason, several food-processing procedures including roasting, microwaving or fermenting may affect the CGA content. For example, Pollini et al. demonstrated that the microwave assistance of *Lycium barbarum* leaf extracts reduced the CGA yield [54], and that water with different mineralization levels produce different isomerization of CGA after microwave treatment [55]. In blueberry jam production, the high-temperature processing of blueberries with sucrose promoted the formation of 11 CGA derivatives [56]. Moreover, the roasting process (170 to 200 °C/10 to 30 min) of coffee beans promotes CGA transformation to four chlorogenic acid lactones [57]. On the other hand, ultrasound-assisted extraction of CGA from potato sprouts has been shown to enhance recovery compared to maceration. This process involves utilizing water content (30%), a time duration of 5 min, and ascorbic acid (1.7 mM) as an anti-browning agent. [58]. Chen et al. evaluated various ultrasound conditions for CGA extraction from burdock (*Arctium lappa* L.) roots. Their study demonstrated selective extraction of CGA and cynarin between frequencies of 40 kHz and 120 kHz [59].

## 4. Biosynthesis of Chlorogenic Acid

The biosynthesis of CGA proceeds through the phenylpropanoid pathway, originating from the amino acids phenylalanine or tyrosine. The identification and functional characterization of genes and enzymes involved in this pathway have been facilitated by knockout mutations and RNAi-mediated suppression in Arabidopsis and other model plants [60,61]. The schematic representation of this pathway is illustrated in Figure 2.

CGA is synthesized in one step by the enzyme hydroxycinnamoyl-CoA: quinate hydroxycinnamoyltransferase (HQT), which promotes the ester binding between 4-Coumaroyl-CoA as the hydroxycinnamoyl donor for the esterification with quinic acid giving CGA and CoA [62,63,64]. Alternatively, 4-Coumaroyl-CoA can be esterified with shikimic and quinic acid by hydroxycinnamoyl-coA shikimate/quinate hydroxycinnamoyl transferase (HCT), giving 4-coumaroylshikimic acid or 4-coumaroylquinic acid, respectively. The 4-coumaroylquinic acid can be m-hydroxylated by p-coumarate 3-hydroxylase (C3H), giving CGA. On the other hand, 4-coumaroylshikimic acid is hydroxylated by C3H giving caffeoyl shikimic acid, then HCT promotes the formation of caffeoyl CoA and shikimic acid, finally, caffeoyl CoA and quinic acid can be esterified by HQT giving CGA [65].

## 5. Absorption, Distribution, and Metabolism of Chlorogenic Acid

CGA (5-caffeoylquinic acid) and dicaffeoylquinic acids are the main CGA in nature, the bioavailability of CGA was studied in 10 healthy adults by Farah et al. [66]. After the intake of a green coffee extract (decaffeinated) containing 170 mg of CGA in the forms of chloroquinic acids (5-CQA, 4-CQA, and 3-CQA), together with diCQA and ferulicquinic acid (FQA), the plasma and urine were analyzed. FQA was not detected in the plasma of any people due to its poor absorption and faster uptake by the liver compared to chloroquinic acids and diCQA [67]. In plasma, three CQA and 3 diCQA were detected; along with caffeic, ferulic, isoferulic, and p-coumaric acids. Meanwhile, in the urine, excretion products included 4-CQA, 5-CQA, sinapic, p-hydroxybenzoic, gallic, vanillic, dihydrocaffeic, caffeic, ferulic, isoferulic, and p-coumaric acids [66].

Metabolism of CGA has been deeply studied in Wistar rats. In summary, the plasma concentration of CGA reached a maximum concentration after 2 h with C max 3.45 nmol/mL, and approximately 50% of the CGA was found in the plasma in free form; while 90% of caffeic acid was found as glucuronated and/or sulfate derivatives [68]. After 4 h, microbial metabolites began to appear, reaching their maximum at 8 h after gavage. The bacterial metabolites were identified as dihydrocaffeic and 3-hydroxyphenylpropionic acids [68], as shown in Figure 3. This is further evidence that urine is the major route of excretion of polyphenols or derivatives thereof, after 4–8 h. Only 3.3–4.0% of total ingested polyphenols were recovered, which were not metabolized [69].

## 6. Toxicological Profile of CGA

Venkatakrishna et al. evaluated the acute and subchronic toxicity of a standardized green coffee bean extract (CGA-7™) that contains 50% chlorogenic acids in Wistar albino rats. A single oral dose of 2000 mg/kg for 14 days and repeated doses for 90 days were tested to evaluate the risk of long-term use of CGA-7™. Observation for 14 days revealed no clinical signs of toxicity or mortality in animals receiving the acute oral dose of 2000 mg/kg of CGA-7™ and biochemical parameter changes were toxicologically insignificant, remaining within the physiological range; repeated dose of 90-day CGA-7™ did not affect the normal metabolism or physiology of the animals up to 1000 mg/kg. Macroscopic and histological examination of the organs revealed no organ toxicity, suggesting the safety of CGA-7™ [70]. A similar study that evaluated the acute and subacute toxicity of the crude aqueous extract of *C. pachystachya* leaves (CGA: 27.2 ± 0.94 mg/g of extract) through an in vivo model, as well as cytotoxicity, genotoxicity, and antioxidant activity in vitro, showed similar results in animals. In a subacute toxicity assay, water and food intake, and organ weights were not markedly different between animals that were treated and negative control. In the biochemical analyses, all values are within physiological levels, the color of the organs appeared normal and the architecture was preserved; and the histopathologic analysis of the liver, kidneys, and stomach indicated architecture organs with normal aspects. However, in vitro evaluations showed cytotoxicity in HT-29 cells (IC50 = 4.43 μg/mL), as well as genotoxic effects from 0.31 to 2.5 mg/mL [71]. This possible effect was also evaluated in the bone marrow of diabetic rats, which showed increased micronuclei formation and a decreased PCE/NCE ratio, indicating possible genotoxic and cytotoxic side effects in bone marrow [72]. It is considered that a decrease in the ratio of polychromatic erythrocytes (PCE) to normochromatic erythrocytes (NCE) (P/N) in the micronucleus test is an indicator of bone marrow toxicity induced by mutagens [73].

Concerning the toxicity profile in humans, a double-blind, placebo-controlled clinical study in which healthy overweight male and female subjects (N = 71) were randomly assigned to receive 500 mg daily of CGA-7™ or placebo for 12 weeks evidenced that individuals receiving CGA-7™ showed no adverse effects, and their biochemical and hematological parameters varied only marginally; this study provided a scientific validation of the functionality and safety of the green coffee bean extract CGA-7™ in healthy individuals [74]. This observation was also confirmed in a randomized, double-blind, placebo-controlled clinical trial (NCT02621060) conducted on 30 patients to evaluate the effect of chlorogenic acid in patients with glucose intolerance: Patients received a dose of 1200 mg per day, distributed in three 400 mg capsules three times a day for 90 days of chlorogenic acid or placebo. Toxicity results showed that only 5% of the participants had adverse effects such as diarrhea (6.6%, 1/15) and abdominal pain (6.6% 1/15). It is worth noting that the consumption of supplements rich in CGA revealed anti-obesity effects [25], in this sense, CGA-7™ causes the increase of lean mass/fat mass ratio and serum lipid profile [74], while Altilix^®^, a supplement containing chlorogenic acid and luteolin, improved hepatic and cardio-metabolic parameters in subjects with metabolic syndrome [75].

## 7. Role of Chlorogenic Acid in Chemoresistance

Drug resistance is acknowledged as the primary reason for the failure of chemotherapy in the treatment of most human tumors. Its prevalence is highly influenced by factors such as the type of cancer, the specific drug used, and the patient’s comorbidities [76]. There are two distinct types of chemoresistance based on their development timeline. Intrinsic resistance occurs before drug administration, rendering tumors unresponsive to chemotherapy from the outset. On the other hand, acquired resistance emerges after an initial positive response to antineoplastic drugs, leading to the proliferation of a resistant cell population and subsequent uncontrolled tumor regrowth [77]. Currently, the basis of chemoresistance is well elucidated [78,79] and summarized in eight mechanisms. First is tumor heterogeneity; second is drug inactivation by cytochrome P450 (CYP), glutathione-S-transferase (GST) and uridine diphosphate-glucuronosyltransferase (UGT) superfamily. CYP enzymes are membrane-bound hemoproteins that play a pivotal role in detoxifying xenobiotics, cellular metabolism, and homeostasis [80]. GSTs catalyze the reaction of the sulfhydryl group of the tripeptide glutathione of various xenobiotics. This sulfhydryl group reacts with electrophilic sites on xenobiotics, forming conjugates that are more readily excreted and typically less toxic than the original drug. In addition to this direct detoxication, GSTs catalyze the secondary metabolism of compounds oxidized by other enzymes [81]. Glucuronidation, facilitated by UGTs, constitutes a significant phase II biotransformation pathway. It serves as a crucial cellular defense mechanism, complementing phase I metabolism and membrane transport. This process is instrumental in the inactivation of therapeutic drugs, various xenobiotics, and endogenous molecules [82]. Third, alterations in drug targets, such as members of the epidermal growth factor receptor (EGFR) family; and targeted downstream signaling partners such as Ras, Src, Raf, and MEK. Fourth, increased expression of efflux pumps, which reduces intracellular drug concentration (MDR or P-gp, MRP1, ABCG2). Fifth, DNA damage repair mechanisms. Sixth, inhibition of cell death is mediated by overexpression of BCL-2 family proteins, Akt, NF-κB, and STAT. Seventh, the promotion of epithelial-mesenchymal transition (EMT) is facilitated by overexpression of factors such as transforming growth factor β (TGFβ), FAK kinase, and vascular endothelial growth factor (VEGF). Finally, epigenetic changes also play a role in resistance to chemotherapeutic drugs [83].

Reversing chemoresistance poses a significant challenge in clinical settings. However, the existing knowledge about the underlying mechanisms has paved the way for the development of strategies aimed at enhancing conventional drug therapy to overcome this hurdle [84]. In cancer treatment, combining therapies with different antineoplastic agents allows for a synergistic effect, increasing the destruction of a larger number of cells and reducing the likelihood of the cancer developing resistance to a specific drug. In this context, the concurrent use of natural products has been under consideration for many years [85], due to their ability to suppress malignant cell survival and proliferation by targeting various molecular pathways and signal transduction mechanisms [86]. When cancer cells are treated with natural products in combination with chemotherapeutic drugs, there is an observed additive cytotoxic effect, attributed to the activation of alternative pathways inducing cell death or prolonging the intracellular presence of drugs [87]. In a systematic review encompassing 110 publications, phenolic derivatives and flavonoids emerged as the main compounds studied as chemo-sensitizers. Notably, curcumin, resveratrol, and epigallocatechin-3-gallate were frequently cited in combined treatments with clinically used chemotherapeutics [87]. Up to the date of publication of the above reviews, there was little evidence about the role of CGA in chemoresistance, and therefore, none of them mentioned these compounds. However, in the last 5 years, a significant amount of research has been published demonstrating their potential as a chemosensitizer in vitro through the regulation of specific signaling pathways that directly or indirectly influence the enhancement of chemo-sensitization to specific drugs of clinical use in cancer, as well as potentiating the effects of conventional drugs in animal models.

Investigating the impact of CGA on resistant human hepatocellular carcinoma cell lines (HepG2 and Hep3B), CGA was found to increase the sensitivity of hepatocellular carcinoma cells to 5-FU treatment [88]. As for other drugs, such as doxorubicin (DOX), CGA was shown to collaborate by significantly reducing cell viability and growth through induction of apoptosis, attributed to inhibition of extracellular signal-regulated kinases (ERKs) [88,89]. ERKs are the culmination of a mitogen-activated protein kinase cascade that regulates cellular processes such as proliferation, migration, and survival. Consequently, abnormal ERK signaling often plays a role in the tumorigenesis and metastasis of numerous cancers [90]. In vivo studies have documented that treatment with CGA and/or DOX results in a substantial decrease in solid tumor volume and weight in the Ehrlich carcinoma model (SEC) in Swiss female albino mice. This treatment up-regulates TRAIL/TRAILR2, FasL/Fas, and caspase-3 (CASP 3) gene expression while down-regulating Bcl-2 gene expression. In addition, there is a marked increase in TRAIL/TRAILR2 gene expression. In addition, there is a marked increase in the expression of active caspase-3 leading to apoptosis [91]. Tumor necrosis factor-related apoptosis-inducing ligand (TRAIL) preferentially induces apoptosis in neoplastic cells by binding to its receptors TRAIL-R1 and TRAIL-R2. This biological principle has been adopted for the development of targeted cancer therapies whereby an increase in the amount or activity of TRAIL-R2 on cancer cells may allow more cells to be destroyed [92]. Fas and Fas Ligand (FasL) are members of the tumor necrosis factor (TNF)-receptor and TNF family, respectively. The ligation of Fas with FasL results in the activation of a caspase cascade that initiates apoptosis [93]. CASP 3 is a member of the cysteine protease family that cleaves its targets at aspartic acid residues and is a key player in the apoptosis pathway [94]. Bcl-2 is a major neuroprotective protein of the Bcl-2 family, which consists of anti-apoptotic (Bcl-2 and Bcl-x) and pro-apoptotic (Bad/Bik, Bax, Bak, tBid, and Bim) proteins, and is localized on the mitochondria membrane, endoplasmic reticulum, and golgi [95]. Furthermore, because CGA is known to influence MAPK and PI3K/Akt signaling pathways [96], combination therapy with Regorafenib has been explored [97], which is a multikinase inhibitor used in the treatment of refractory advanced colon cancer [98]. Refolo et al. investigated the impact of CGA in hepatocarcinoma cells (PLC/PRF/5), finding that the combined treatment reduced cell growth and enhanced apoptosis by enhancing the inhibition of MAPK and PI3K/Akt/mTORC pathways. In this regard, Regorafenib is known to inhibit the mitogen-activated protein kinase (MAPK) cascade but has a more modest effect on the PI3K/Akt signaling pathway [99], therefore CGA potentiates its effect by complementing the inhibition of this key signaling pathway in the pathogenesis of advanced colon cancer [97]. Both MAPK (mitogen-activated kinases) and PI3K/Akt/mTOR (phosphatidylinositol 3-kinase/protein kinase B/ rapamycin target in T cells) are important intracellular signaling pathways that regulate a variety of cellular processes, including survival, proliferation, differentiation, metastasis, and angiogenesis [100]. The summary of these investigations reported since 2015 is shown in Table 2.

## 8. Protective Role of Chlorogenic Acid against Toxicity Induced by Chemotherapy

The first mention of CGA as a possible chemoprotectant was made in 1993 by Abraham et al. After that, this property of CGA was not studied again until 2004, when numerous studies were carried out on the protective capacity of CGA against chemotherapy [102,103]. Figure 4 summarizes the main protective effects of CGA against toxicity induced by chemotherapy.

Typically, neoplastic diseases are treated with the maximum tolerable dose to achieve the highest possible cell death percentage [104]. Cyclophosphamide is employed in the treatment of chronic and acute leukemias, multiple myeloma, lymphoma, retinoblastoma, neuroblastoma, breast cancer, and ovarian cancer. It is also a component of primary induction regimens preceding bone marrow transplantation. However, its clinical application is frequently limited by side effects and toxicities encompassing bone marrow suppression, nephrotoxicity, hepatotoxicity, ovotoxicity, urotoxicity, immunotoxicity, cardiotoxicity, mutagenicity, teratogenicity, and carcinogenicity [105]. Studies have explored molecules that confer protection against cyclophosphamide-induced toxicity. Alkis et al. found that CGA has a protective effect on cyclophosphamide-induced ovotoxicity in albino female Wistar rats. The study involved a single dose of cyclophosphamide (200 mg/kg body weight) on the first day, followed by an oral dose of CGA (100 mg/kg/d) for 7 days. The results indicated a reduction in oxidative stress in ovarian tissue, as demonstrated by the decrease in 8-hydroxy-2-deoxyguanosine (8-OHdG), which is a major product of DNA oxidation [106]. CGA was able to diminish cyclophosphamide-induced histological damage in rat ovaries, maintaining a follicle count close to normal. CGA reduced necrosis in many luteal cells, severe hyperemia and hemorrhage in vessels, degeneration in germ cells, follicles dominated by luteal structures, and primordial follicles [107].

Doxorubicin or adriamycin are the most used type of cytotoxic antibiotics that act on DNA; they are natural substances produced by fungi that are capable of altering cell growth. DOX-induced cardiotoxicity results from various factors, including increased production of reactive oxygen species (ROS), release of inflammatory mediators, and disrupted intracellular calcium cycling homeostasis in the heart. The chemoprotective effect of CGA against DOX-induced toxicity in cardiomyocytes of male Wistar rats was studied in 2004. In this study, DOX-induced lipid peroxidation in heart membranes, mitochondria, and microsomes was estimated. The protective capacity of this acid was compared with dexrazoxane, which is used as an adjuvant during DOX chemotherapy. After preincubation of cardiomyocytes with the test compound (200 µM; 1 h), the cardiomyocytes were treated with the toxic agent DOX (100 µM; 8 h). CGA was non-cytotoxic and stabilized both the membranes and the energetic state of cardiomyocytes. The test compound protected cardiomyocytes against DOX-induced oxidative stress in all monitored parameters and was more effective than dexrazoxane. CGA significantly reduced DOX-induced lipid peroxidation of cardiac membranes, being the most effective compound of the tested compounds in this field [103]. Subsequently, in 2019, Abd et al. reported the cardioprotective effect of CGA against doxorubicin-induced cardiotoxicity in female Swiss albino mice. The dose used was 60 mg/kg/d orally for 21 days; this dose was administered two hours before injecting DOX at a concentration of 2 mg/kg/day for the same 21 days. According to this study, CGA significantly ameliorated the oxidative stress caused by DOX, as evidenced by reduced malondialdehyde (MDA) levels and increased glutathione (GSH) levels [91]. MDA is the most frequently used biomarker of oxidative stress in many health problems such as cancer. MDA is one of the end products of lipid peroxidation, which is a highly toxic compound that crosslinks cellular macromolecules, such as proteins and DNA, conferring antigenic properties [108]. GSH, a ubiquitous and redox-active small molecule, plays a critical role in cellular and organismal health. These findings align with a notable reduction in serum creatine kinase-myoglobin binding (CK-MB) activity and serum lactate dehydrogenase (LDH) levels, both being common biomarkers of cardiac damage. Additionally, the administration of CGA led to a decrease in vacuolization of cardiac myocytes induced by DOX and inflammation [91]. The cardioprotective role of CGA against doxorubicin was also recently studied in H9c2 rat cardiomyocyte cells; it was reported that CGA enhanced cell growth rate by approximately 27% and greatly counteracted doxorubicin-induced cell death from 12.65% to 6.08% [89].

The effective protection against cardiac membrane peroxidation can be attributed to the antioxidant properties of CGA, as it has been shown to protect cardiomyocytes from necrosis and apoptosis during H_2_O_2_-induced injury, by inhibiting ROS generation and activation of the caspase-3 apoptotic pathway [109].

Antimetabolites are drugs that interfere with nucleic acid synthesis and act mainly on fast-growing tumors [110]. Some of them are pyrimidine analogs, purine analogs, and antifolates. Pyrimidine analogs, such as cytarabine (Ara-C) and 5-FU, constitute an important pharmacological group in antineoplastic treatment, with broad-spectrum activity [111]. 5-FU is a fluorinated pyrimidine analog of uracil that acts by competitive inhibition of thymidylate synthase. In addition, it is incorporated into RNA and DNA, and alters their function [88]. It is mainly used for adjuvant treatment of breast cancer, gastric cancer, and colorectal cancer. It is also used in the first-line treatment of head and neck neoplasms and gastrointestinal tumors [112,113,114]. Although it is a potent anticancer agent, its use is limited by its significant tissue toxicity that is associated with increased oxidative stress and inflammation [19,115]. The generation of free radicals leading to cell membrane damage and 5-FU-induced lipid peroxidation are considered to be the main mechanisms underlying its toxic effects [116]. Its most frequent secondary effects are cardiotoxicity, nephrotoxicity, myelosuppression, and palmoplantar syndrome [22].

The efficacy of CGA against 5-FU-induced nephrotoxicity has been documented by Rashid et al. in 2016. In their study, Wistar rats received daily oral administration of CGA at a concentration of 120 mg/kg for 20 consecutive days. On day 19, a single intraperitoneal injection of 150 mg/kg of 5-FU was administered. CGA demonstrated a significant reduction in 5-FU-induced lipid peroxidation (LPO), lowered serum toxicity markers, and restored the antioxidant balance, effectively mitigating apoptotic and inflammatory tissue damage induced by 5-FU. The histological findings supported the biochemical and immunohistochemical results obtained [22]. Additionally, Mentese et al. recently reported the chemoprotective role of CGA against 5-FU-induced ovarian tissue toxicity. Their research suggests that CGA acts as a protective agent by inhibiting oxidative stress and inflammation in experimental models [20]. The study revealed that systemic administration of 5-FU (3 mg/kg/d) orally for 3 days suppressed the activity of catalase (CAT) in ovarian tissue—an enzyme responsible for catalyzing the reduction of hydrogen peroxide to water [117]—and increased MDA levels. However, CGA treatments led to a significant and dose-dependent increase in CAT activity, effectively reducing MDA levels. This was accompanied by a decrease in cumulative total oxidant status (TOS) and an increase in overall ROS scavenging capacity by increasing total antioxidant status (TAS). TOS is usually used to evaluate the overall oxidation state of the body, while the TAS is applied to measure the overall antioxidant status [118]. The administration of CGA also resulted in decreased DNA damage, indicated by a significant reduction in 8OHdG levels in rat ovarian tissue. Notably, CGA in combination with 5-FU was able to reduce the levels of interleukin 6 (IL-6)—a critical cytokine involved in the proinflammatory process—which is elevated in rats exposed only to 5-FU and associated with ovarian tissue damage [20].

Antifolates are compounds analogous to folic acid; the most widely used is methotrexate (MTX), which has a structural similarity to dihydrofolic acid. It binds to dihydrofolate reductase, thereby inhibiting the step from dihydrofolate to tetrahydrofolate, which acts as a donor of monocarbon groups for the synthesis of purines and pyrimidines, and therefore of nucleic acids [119]. Its antitumor activity is wide ranging; it is used in the treatment of various neoplasms, and plays a crucial role as an adjuvant in the treatment of breast cancer and leukemias. However, its use can lead to testicular toxicity, hepatotoxicity, myelotoxicity, and mucositis [120]. The protective effect of CGA against MTX-induced testicular damage in male Wistar rats has been reported. The results obtained show that in rats treated with CGA, the number of detached, atrophic, and degenerated seminiferous tubules was statistically lower than in the group treated with MTX alone. The use of CGA decreased CASP-3 levels and attenuated the MTX-induced increase in MDA levels. The use of CGA caused an increase in CAT, SOD, and GSH-Px activity which were decreased by MTX, obtaining parameters similar to the healthy control group [121]. Subsequently, the same group of researchers reported the protective effect of CGA on methotrexate-induced cerebellar Purkinje cell damage in male Wistar albino rats. They found that CGA reduced oxidative stress by increasing the activity of SOD, CAT, and GSH. This resulted in significantly reduced Purkinje cell damage and apoptotic cell expression in the cerebellum of MTX-treated rats [121]. The use of CGA was also reported as a protective agent against MTX-induced hepatotoxicity. In this study, doses of 100 mg/kg/d of CGA were administered orally for 20 days, and it was found that CGA improved MTX-induced histology and decreased oxidative stress. In addition, CGA inhibited inflammation and apoptosis mediated by COX-2, inducible nitric oxide synthase (iNOS), BAX, Bcl-2, CASP-3, and Caspase 9 (CASP-9) [120]. iNOS is involved in immune response, binds calmodulin at physiologically relevant concentrations, and produces nitric oxide radical (NO) as an immune defense [122]. CASP9 is a well-known initiator caspase that triggers intrinsic apoptosis [123]. Regarding neoplastic purine analogues, such as 6-mercaptopurine, azathioprine, 6-thioguanine, and allopurinol—whose damages are mainly hematological, gastrointestinal and hepatic—no literature was found to support the use of CGA to prevent their harmful effects, which is why it is an interesting field of study to address.

The platinum derivatives constitute another crucial group in cancer treatment, particularly those impacting cellular DNA. Notably, cisplatin and oxaliplatin are among the extensively used members of this group. Cisplatin or CIS-diaminodichloroplatinum II (CP) stands out as a paramount antineoplastic drug, exerting its cytotoxic effect by alkylation of the DNA double helix. This process results in intercatenary adducts, inhibiting cell division and demonstrating heightened efficacy in rapidly replicating cells [124]. At elevated doses, CP induces acute apoptosis through oxidative stress and inflammation induction [17,125,126]. Its wide-ranging applications include the treatment of solid tumors such as head–neck, breast, ovarian, bladder, cervix, testis, small and non-small cell lung (NSCLC), and liver metastases [127,128]. However, CP chemotherapy is associated with severe nephrotoxicity and renal dysfunction, constituting one of its most pronounced dose-limiting side effects [15]. CP toxicity has been linked to increased activity in CASP-3, caspase-8 (CASP-8), and CASP-9, cytochrome c release, apoptosis-inducing factor (AIF) translocation, ROS generation, and NF-κB activation [129]. Additional side effects stemming from CP therapy encompass ototoxicity, gastrointestinal toxicity, myelosuppression, ovarian toxicity, and neurotoxicity [15]. Nephrotoxicity and renal dysfunction significantly limit the use of CP in treatment. Domitrovic et al. studied the protective effects of CGA against renal damage caused by CP, finding that the use of CGA at a dose of 30 mg/kg/d for two consecutive days attenuates CP-induced renal injury by suppressing oxidative stress and inflammation, and enhancing renal regeneration in rats. CGA was found to suppress the expression of p53, active CASP-3, BAX, cyclin D1 and microtubule-associated protein 1 light chain 3 isoform B (LC3B), achieving reduced apoptosis and autophagy in renal tissue. In addition, it was determined that CGA decreases the activity of heme oxygenase (HO-1) and cytochrome P450 2E1 (CYP450E1) enzymes. The latter is related to drug metabolization. The down-regulation of TNF-α and COX-2 indicates that CGA also managed to reduce inflammation levels in renal tissues [15]. Recently, CGA has been shown to intervene by down-regulating TLR4/NLPR3/IL-1β signaling pathways and inhibiting Caspase-1/GSDMD signaling; this was demonstrated by Badr et al., who administered a dose of 20 mg/kg/d, for 14 days to Wistar mice. The use of CGA significantly attenuated CP-induced oxidative stress in the kidneys, decreasing lipid peroxidation levels and GPx activity, normalizing MDA and antioxidant levels, and increasing catalase activity. In addition, it succeeded in reducing inflammatory responses to CP in terms of NF-κB and TNF-α. This was reflected in a significant improvement in terms of CP damage in renal tissue histology, showing almost intact morphological features of the tubular epithelium, with minimal evidence of dilated or degenerated tubular segments [17].

Another problem associated with the use of CP is ovarian failure in up to 40% of patients and hormonal imbalance, which may result in temporary or permanent infertility [130]. The use of CP often causes ovarian failure, increasing follicular apoptosis, reducing follicle production, and altering the reproductive cycle [131]. The protective role of CGA on CP-induced ovarian damage has been reported in recent years. It was shown that the use of CGA at a dose of 3 mg/kg/d for 3 days after a single dose of CP (5 mg/kg), improved ovarian tissue histology and significantly reduced the levels of markers of oxidative stress, inflammation, and apoptosis in a dose-dependent manner [132].

A final subgroup within the category of drugs affecting cellular DNA consists of those derived from camptothecins, which are natural compounds originating from camptothecin, an alkaloid found in the Chinese tree Camptotheca acuminata [133]. These drugs disrupt the normal progression of DNA replication and RNA synthesis by selectively and irreversibly inhibiting topoisomerase type I [134]. With its broad spectrum of antitumor activity, irinotecan is commonly prescribed for metastatic colon cancer, cervical cancer, and small-cell lung carcinoma; while topotecan is utilized for ovarian cancer and small-cell lung cancer. However, both medications frequently result in myelosuppression, hepatotoxicity, and mucositis [135]. Despite the wide use of irinotecan and topotecan, studies on the hepatoprotective effects of CGA against the toxicities induced by camptothecins are lacking in the existing literature. Further exploration in this area is warranted to comprehensively understand the potential protective role of CGA. In the realm of antineoplastic drugs of natural origin, they typically target processes such as cell meiosis rather than directly impacting DNA [136]. Triptolide (TP), a diterpenoid epoxide derived from *Tripterygium wilfordii*—a Chinese medicinal herb—possesses notable antitumor, immunosuppressive, and anti-inflammatory properties. However, its clinical utility is constrained by low solubility, limited bioavailability, and notable toxicity; particularly hepatotoxicity [137]. A study in Krushinsky-Molodkina (KM) rats demonstrated the potential of CGA as a hepatoprotectant against TP-induced hepatotoxicity. Rats treated with CGA (40 mg/kg/d) for 7 consecutive days and then injected intraperitoneally with a single dose of TP (1 mg/kg) exhibited reduced oxidative stress, increased Nuclear factor erythroid 2-related factor 2 (Nrf2) accumulation in the nucleus, and improved liver function markers. Nrf2 is a transcription factor that regulates the cellular defense against toxic and oxidative insults through the expression of genes involved in oxidative stress response and drug detoxification. CGA significantly mitigated TP-induced elevation of serum alanine transaminase, aspartate transaminase, and hepatic MDA. Furthermore, it elevated hepatic GSH, glutathione S-transferase (GST), glutathione peroxidase (GPx), SOD, and CAT levels [138]. GST and GPx are essential components of cellular detoxification systems [139]. It has been shown that the antioxidant activity of CGA depends on the activation of Nrf2-dependent or Nrf2-independent pathways. The antioxidant properties of CGA are high and have been evaluated for its free radical scavenging effect by 2,2-diphenyl-1-picrylhydrazyl (DPPH) radical and lipid oxidation prevention activity by Rancimat assay. The results showed that the DPPH radical scavenging activity is higher in CGA than in α-tocopherol and butylated hydroxytoluene (BHT), with values of 36.3, 32.5, and 8.9%; while the antioxidant index (in corn oil) showed the same pattern CGA > α-tocopherol > BHT, with values of 2.33, 1.93, and 1.72, respectively [140]. It has also been demonstrated that CGA can scavenge radicals present in our bodies like hydroxyl radical (·OH) and nitric oxide radical (NO) [141,142].

Tunicamycin (Tun)—a natural antibiotic from *Streptomyces lysosuperificus*—blocks N-linked glycosylation in proteins, inducing endoplasmic reticulum (ER) stress and triggering apoptosis and autophagy [143]. Its potential in malignancy treatment, especially in triple-negative breast cancer (TNBC), has been explored due to its ability to induce ER stress-related autophagy, apoptosis, and chemosensitization through the PI3K/AKT/mTOR signaling pathway [144]. However, Tun’s high toxicity, causing neuronal cell death, acute liver and kidney injury, and testicular damage, hinders its clinical application [145]. A recent study investigated CGA’s protective role against testicular inflammation and apoptosis induced by Tun-induced ER stress. CGA administration significantly reduced the expression of inflammatory and apoptotic genes, alleviated structural changes in seminiferous tubules, and mitigated biochemical markers associated with testicular damage [146].

Among the drugs that act on extracellular cell division factors, such as estrogen and androgen antagonists, tamoxifen (TAM) is the most widely used anti-estrogen for the treatment of estrogen receptor-positive breast cancer. Its pharmacological activity is known to depend on its conversion to its active metabolite, endoxifen, by CYP2D6 [147]. However, among the possible adverse effects of prolonged use of tamoxifen we can find hepatotoxicity, nephrotoxicity, reproductive problems, and ocular toxicity [148,149,150]. These effects are believed to be related to the overproduction of oxygen radicals that occurs during the metabolic activation of TAM [151]. The protective use of CGA was recently reported in two parallel studies by Owumi et al. In the first study, the protective role of CGA against TAM-mediated reproductive toxicities was studied in male Wistar rats. The result showed that co-treatment of TAM (50 mg /kg/d) and CGA (50 mg/kg/d) for 14 days significantly increased testosterone, luteinizing hormone (LH), and the follicle-stimulating hormone (FSH) levels, and decreased prolactin level compared to the group of rats treated with TAM alone. CGA reduced the decrease of acid phosphatase, alkaline phosphatase and antioxidant enzymes in the testis, and alleviated the increases of ROS and reactive nitrogen species (RNS), myeloperoxidase (MPO), nitric oxide, IL-1β, and TNF-α in the epididymis and testis of TAM-treated rats. In addition, CGA increased anti-inflammatory cytokines IL-10, suppressed CASP-3 activity, and reduced pathological lesions in the examined organs of co-treated rats. Finally, the phytoprotective effect of CGA improved the reproductive function of rats affected by TAM-mediated toxicities by reducing oxido-inflammatory damage and apoptotic responses [150]. In a subsequent study by the same researchers, CGA was also observed to diminish oxido-inflammatory and apoptotic responses in the liver and kidney of TAM-treated rats. Among their notable findings, CGA was shown to elevate hepatic and renal GPx, GST, and GSH activity, as well as SOD, CAT, and TSH activity. Furthermore, CGA down-regulated IL-1β and TNF-α, while increasing IL-10 levels [149]. No reports have been found studying the use of CGA as protective against toxicity caused by other estrogen or androgen antagonists. There are also no studies on the use of CGA against side effects caused by antineoplastic drugs that act on the immune system, such as monoclonal antibodies.

## 9. Adjuvant Role of Chlorogenic Acid in Radiotherapy

Radiotherapy, employing ionizing radiation (X or Y), stands out as a highly effective cytotoxic therapy for localized solid cancers. It is utilized in over 60% of cancer cases for both curative and palliative purposes, either as a standalone treatment or in conjunction with other oncological interventions [152,153]. Despite its effectiveness, radiation treatment poses challenges by damaging or destroying normal cells, particularly skin cells, leading to severe dermatological issues and even carcinogenesis [154]. Additional side effects encompass testicular tissue damage during pelvic cancer treatment, DNA damage to blood lymphocytes, and chromosomal damage to healthy cells overall [102,155,156]. While advanced radiotherapy techniques such as intensity-modulated radiotherapy aim to minimize toxicity, some patients still encounter adverse effects [153]. The radioprotective potential of CGA against γ-radiation-induced chromosomal damage in male albino Swiss mice was initially demonstrated in 1993. Results indicated that a single oral administration of CGA (200 mg/kg) either 2 h before or immediately after irradiation significantly reduced micronucleated polychromatic erythrocyte frequencies (Mn PCE) induced by whole-body exposure to γ-radiation [102]. This protective effect was further investigated in a study in 2013, which focused on CGA’s role in shielding human blood lymphocytes in vitro from X-ray-induced DNA damage. The study found that CGA reduced radiation-induced DNA damage by 4.49–48.15%, as determined by the alkaline comet assay [155]. In a recent study, the radioprotective potential of CGA against ionizing radiation-induced testicular toxicity was investigated. The findings demonstrated CGA’s ability to attenuate extensive damage to spermatogenesis, seminiferous tubules, basal lamina, Leydig cells, and sperm parameters induced by irradiation. Additionally, CGA restored MDA levels to near-normal, significantly increased SOD, total antioxidant capacity (TAC), and GSH levels, while significantly reducing the ratio of BAX/Bcl-2 compared to the radiation-only group [156].

The antibiotic vancomycin (VCM) has been used in patients undergoing radiotherapy because it alters the gut microbiome in a way that may help prepare the immune system to attack tumor cells more effectively after radiotherapy [157]. VCM is the drug of choice for treating methicillin-resistant *Staphylococcus aureus* (MRSA), but it has been associated with significant nephrotoxicity [158]. The protective effect of CGA on VCM-induced nephrotoxicity in male Wistar rats was investigated. Administration of VCM led to significantly elevated levels of blood urea nitrogen and serum creatinine. However, co-administration of CGA prevented these increases. CGA also effectively reduced VCM-induced oxidative stress, as evidenced by decreased levels of MDA and NO in the kidneys. Additionally, CGA administration prevented the decline in renal antioxidative enzyme activities, such as glutathione reductase (GSR), GPx, and CAT, and restored depleted levels of reduced GSH caused by VCM. Furthermore, CGA administration inhibited VCM-induced expression of NF-kB, inducible nitric oxide synthase, and downstream pro-inflammatory mediators such as TNF-α, IL-1β, and IL-6. Markers of apoptosis were also significantly reduced with CGA treatment. In summary, CGA treatment alleviated the oxidative and nitrosative stresses induced by VCM and counteracted the apoptotic and inflammatory effects of VCM [159].

## 10. Conclusions

CGA, a naturally occurring polyphenol, is safe, orally bioavailable, and exhibits anti-inflammatory, antioxidant, and anticancer properties, making it a promising candidate for cancer prevention and treatment. The data summarized in this review indicate that CGA could serve as a valuable adjunct in both chemotherapy and radiotherapy, mitigating chemoresistance and shielding normal tissues from treatment-related side effects. Today, new formulations are in development for specific delivery of CGA in tumors; for example, by pectin nanoparticles on colon cancer [160], sialic acid-modified liposomes by recognizing Siglec-1 receptor on tumor-associated macrophages [161], chitosan nanoparticles for skin carcinogenesis treatment [162], or albumin-CGA nanoparticles [163]. CGA-nanocarriers loaded with chemotherapeutic drugs would also provide a more specific cancer treatment, with a better prognosis for patients.

Despite the positive effects of CGA in combinatory treatments with anticancer drugs, by in vitro and in vivo, the use of this compound needs to be studied more extensively by clinical trials.

## Figures and Tables

**Figure 1 ijms-25-05189-f001:**
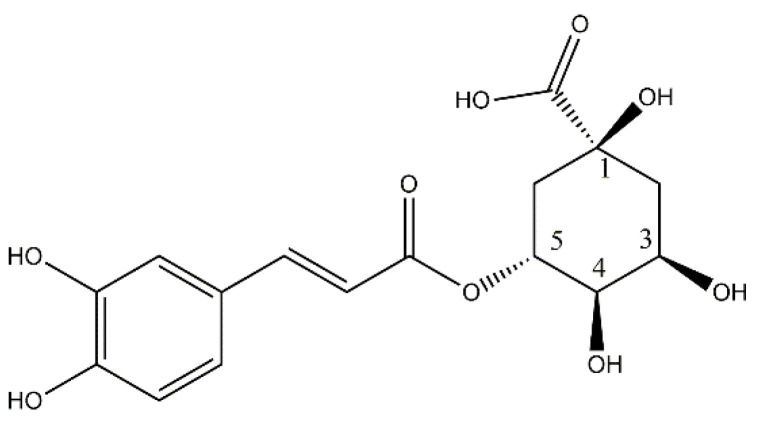
Structure of chlorogenic acid or 5-caffeoylquinic acid (CGA).

**Figure 2 ijms-25-05189-f002:**
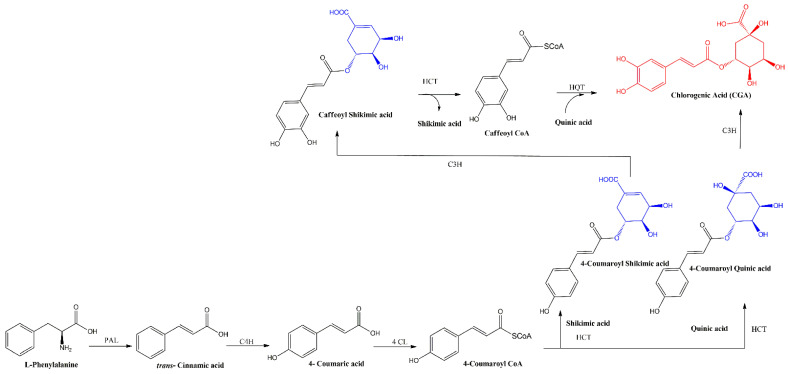
Biosynthetic pathways for the formation of CGA, demarked in red. The involved enzymes are abbreviated: PAL = phenylalanine ammonia lyase; C4H = cinnamate 4-hydroxilase; 4CL = 4-coumaric acid CoA-ligase; HCT = hydroxycinnamoyl-coA shikimate/quinate hydroxycinnamoyl transferase; C3H = *p*-coumarate 3-hydroxylase.

**Figure 3 ijms-25-05189-f003:**
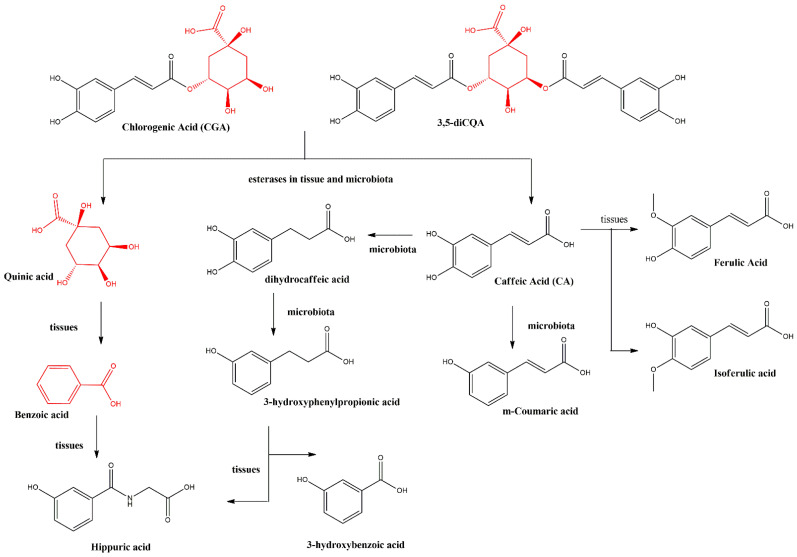
Scheme of biotransformations occurring on CGA and 3,5-diCQA in tissue and by bacterial metabolism [68].

**Figure 4 ijms-25-05189-f004:**
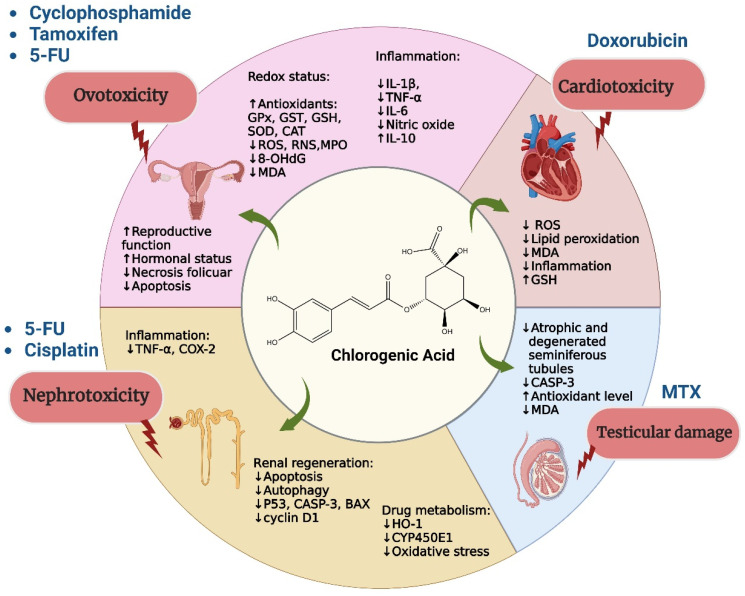
Chlorogenic acid induces protection in healthy organs and tissues affected by chemotherapeutic drugs.

**Table 1 ijms-25-05189-t001:** Chlorogenic acid composition in vegetables.

Plant Source	CGA (mg/kg in dw)	References
Instant coffee	2650–11,600	[26]
Mate tea	4800–24,900	[27]
Sunflower seeds	630–970	[28]
Sweet potato leaves	9600	[29]
English potato ^1^	3.3–9	[30]
Okra ^1^	3.9–21.6	[30]
Eggplant	4980–8050	[31]
Carrot	300–18,800	[32]
Tomato	200–400	[33]

^1^ Results expressed as a wet basis.

**Table 2 ijms-25-05189-t002:** CGA increases the activity of current chemotherapeutic drugs.

Aim	Treatment Conditions	Finding and Mechanisms of Action	References
To explore the influence of CGA on the effects of 5-fluorouracil (5-FU) on human hepatocellular carcinoma cells (HepG2 and Hep3B).	250 μM CGA and 20 μM 5-FU for 4 h	CGA sensitizes hepatocellular carcinoma cells to 5-FU treatment by the suppression of ERK activation through the overproduction of ROS	[88]
To evaluate the antiproliferative effect of CGA in combination with cisplatin or oxaliplatin in cisplatin-sensitive (A431) human cervical carcinoma cell lines.	10^−6^–10^−4^ M CGA and 1 μM cisplatin or 2 μM oxaliplatin for 24 h	Co-treatment with CGA at higher concentrations increased cisplatin activity compared to cisplatin alone. Conversely, lower concentrations of CGA enhanced the activity of oxaliplatin compared to the drug alone.	[101]
To examine the cooperating effects between CGA and doxorubicin (DOX) in U2OS and MG-63 human OS cells.	200 μM CGA and 0.1 μM DOX for 48 h	Concomitant administration of CGA decreased cell viability and growth, promoting cell death potentially via apoptosis induction. CGA + doxorubicin caused a longer-lasting reduction in clonogenic potential. CGA increased inhibition of The ERK1/2 mitogen-activated protein kinase.	[89]
Evaluation of the effects of combined treatment using both low regorafenib concentrations and CGA as a natural compound in PLC/PRF/5 and HepG2 human (hepatocarcinoma cells).	0.1 µM (PLC/PRF/5) or 1 µM (HepG2) and 100 µM of CGA	CGA enhanced regorafenib-mediated cell growth inhibition. CGA potentiated the apoptotic effect of regorafenib by the activation of the pro-apoptotic Annexin V, Bax and Caspase 3/7 and the inhibition of anti-apoptotic Bcl2 and Bcl-xL by inhibition of MAPK and PI3K/Akt/mTORC pathway. Combined treatments were also effective in inhibiting cell motility.	[97]
To evaluate the antitumor effect of CGA alone and in combination with doxorubicin in a solid Ehrlich carcinoma (SEC) model in mice.	CGA (60 mg/kg suspended in 0.5% CMC orally) and DOX (2 mg/kg/day, intraperitoneal) daily for 21 days.	CGA and/or DOX treatment showed a remarkable decrease in solid tumor volume and weight. CGA and/or DOX groups revealed upregulation in gene expressions of TRAIL/TRAILR2, FasL/Fas and caspase-3 genes and down-regulation of Bcl-2 gene expression.	[91]

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
