# Peer review of "Therapeutic Potential of Chlorogenic Acid in Chemoresistance and Chemoprotection in Cancer Treatment"

_ijms, 2024, doi:10.3390/ijms25105189_

Round 1

Reviewer 1 Report

Comments and Suggestions for Authors

The manuscript is remarkable and is expected to attract readers' interest. However, specific aspects need to be addressed and revised prior to publication.

Title

The use of the word “reversal” in that position makes the readers confused that CGA reverses both chemoresistance and chemoprotection.

Abstract

Line 19 - The phrase "like coffee, propolis, yerba mate, eggplant, and medicinal herbs" is redundant since it has already been mentioned that CGA is found in various plants and vegetables.

Line 22 - The sentence "Upon consumption, CGA is rapidly absorbed in the stomach and intestine, and has metabolized by tissues or microbiota" is unclear and could be rephrased for better clarity.

The authors are suggested to briefly explain the mode of action of CGA related to cancer treatment.

Although the abstract does mention its objectives, it does not explicitly highlight any novel findings or gaps in current knowledge. Adding a sentence that identifies specific areas where additional research is urgently needed would enhance the impact of the conclusion. 

Introduction

The authors have to introduce an overview of cancers and current therapeutic treatments. 

The last paragraph of the introduction should be revised. There is a lack of strong rationale stating why this bioactive compound needs to undergo this review linked with previous studies.

Methodology

This review does not have a material and methods section.  Providing elaboration on inclusion and exclusion criteria, search terminologies utilized, and potential biases encountered during the selection process would enhance the methodology of the study. 

Results/Discussion

Section: Functional foods high in chlorogenic acid

As CGA mentioned, the extraction and isolation of CGA derived from coffee and Mate tea should be briefly mentioned.

Section: Biosynthesis of chlorogenic acid

According to Figure 2, the authors must improve the resolution of that figure. Revise “chlonogenic acid”. 

Section: Absorption, distribution, and metabolism of chlorogenic acid

 In this section, the authors did not mention 3,5-dicaffeoylquinic acid. So, it is advisable to include this compound related to metabolism. 

In Figure 3, the authors must improve the figure resolution and carefully check the spelling of compound names/structures such as chlorogenic acid and hippuric acid. 

Section: Rol of chlorogenic acid in chemoresistance

Line 166 – revise Rol to “Role”

Line 218-224 – The statement mentions that CGA influences different pathways, including MAPK and PI3K/Akt, is not clear and mentions only that of Regorafenib.

According to Table 2, What is the 10-6-10-4 M CGA? And where is “N.D.” shown in this table?

The authors are suggested to add the toxicology profile of CGA.

 Section: Protective role of chlorogenic acid against toxicity induced by chemotherapy

The authors discussed the antioxidant potential of CGA by inhibiting DPPH radicals. In reality, these DPPH radicals do not occur in our bodies. So, the authors should additionally discuss some free radicals generated in the body, such as nitric oxide (NO) radicals.

Are there any limitations of this current review? 

 Conclusion

It is suggested that the authors mention future research on CGA that will be focused on overcoming chemoresistance and enhancing chemoprotection in cancer treatment.

References

Please check the style related to the journal, in particular the journal abbreviation.

Comments on the Quality of English Language

Some typos and grammatical errors should be corrected.

Author Response

Response to Reviewer 1.

Title

The use of the word “reversal” in that position makes the readers confused that CGA reverses both chemoresistance and chemoprotection.

Response: Yes, I agree with the reviewer, then I delate the word

Abstract

Line 19 - The phrase "like coffee, propolis, yerba mate, eggplant, and medicinal herbs" is redundant since it has already been mentioned that CGA is found in various plants and vegetables.

Response: It was removed

Line 22 - The sentence "Upon consumption, CGA is rapidly absorbed in the stomach and intestine, and has metabolized by tissues or microbiota" is unclear and could be rephrased for better clarity.

Response: this paragraph was deleted because has no connection with the main idea of the review.

The authors are suggested to briefly explain the mode of action of CGA related to cancer treatment.

Although the abstract does mention its objectives, it does not explicitly highlight any novel findings or gaps in current knowledge. Adding a sentence that identifies specific areas where additional research is urgently needed would enhance the impact of the conclusion. 

Response: the Abstract was improved

Introduction

The authors have to introduce an overview of cancers and current therapeutic treatments. 

Response: This overview was added

The last paragraph of the introduction should be revised. There is a lack of strong rationale stating why this bioactive compound needs to undergo this review linked with previous studies.

Response: The last paragraph of the introduction was modified.

Methodology

This review does not have a material and methods section.  Providing elaboration on inclusion and exclusion criteria, search terminologies utilized, and potential biases encountered during the selection process would enhance the methodology of the study. 

Response: A methodology section was added to the review. The databases used, details on inclusion and exclusion criteria, the main search terminologies used were explained.

Results/Discussion

Section: Functional foods high in chlorogenic acid

As CGA mentioned, the extraction and isolation of CGA derived from coffee and Mate tea should be briefly mentioned.

Response: new information was added in this part.

Section: Biosynthesis of chlorogenic acid

According to Figure 2, the authors must improve the resolution of that figure. Revise “chlonogenic acid”. 

Response: The Figure 2 was edited, and the name was corrected.

Section: Absorption, distribution, and metabolism of chlorogenic acid

 In this section, the authors did not mention 3,5-dicaffeoylquinic acid. So, it is advisable to include this compound related to metabolism. 

Response: This part was improved with more information.

In Figure 3, the authors must improve the figure resolution and carefully check the spelling of compound names/structures such as chlorogenic acid and hippuric acid. 

Response: it was corrected.

Section: Rol of chlorogenic acid in chemoresistance

Line 166 – revise Rol to “Role”

Response: it was check it

Line 218-224 – The statement mentions that CGA influences different pathways, including MAPK and PI3K/Akt, is not clear and mentions only that of Regorafenib.

Response: The paragraph was rewritten

According to Table 2, What is the 10-6-10-4 M CGA? And where is “N.D.” shown in this table?

Response: It was correct it

The authors are suggested to add the toxicology profile of CGA.

Response: A section on the toxicological profile of CGA in animals and humans was added

 Section: Protective role of chlorogenic acid against toxicity induced by chemotherapy

The authors discussed the antioxidant potential of CGA by inhibiting DPPH radicals. In reality, these DPPH radicals do not occur in our bodies. So, the authors should additionally discuss some free radicals generated in the body, such as nitric oxide (NO) radicals.

Response: It was added more information about OH and NO radicals.

Are there any limitations of this current review? 

Conclusion

It is suggested that the authors mention future research on CGA that will be focused on overcoming chemoresistance and enhancing chemoprotection in cancer treatment.

Response: The conclusion was modified

References

Please check the style related to the journal, in particular the journal abbreviation.

Response: The bibliography was added using End Note and the format that was downloaded and applied is specific to that journal.

Reviewer 2 Report

Comments and Suggestions for Authors

Dear Authors

The current review is different from other articles in the field of therapeutic effects of herbal antioxidants. This is because this review has focused on the actual effects of plant antioxidants, which are the cleansing effects of free radicals caused by chemical drugs, not their anti-cancer effects, which are not very actual, because despite many articles, but still the  commercial anti-cancer drugs has not been marketed for them. Therefore, the novelty and initiative of this article is high compared to similar articles in the field of Chlorogenic Acid. For these reasons, I suggest this article to be published in the valuable journal of IJMS. However, as a minor comment, I suggest that the authors do a wider search and if there are other articles in this field, add their results to increase the number of references.

Comments on the Quality of English Language

Minor editing of English language required

Author Response

Response to Reviewer 2.

Comments and Suggestions for Authors

Dear Authors

The current review is different from other articles in the field of therapeutic effects of herbal antioxidants. This is because this review has focused on the actual effects of plant antioxidants, which are the cleansing effects of free radicals caused by chemical drugs, not their anti-cancer effects, which are not very actual, because despite many articles, but still the  commercial anti-cancer drugs has not been marketed for them. Therefore, the novelty and initiative of this article is high compared to similar articles in the field of Chlorogenic Acid. For these reasons, I suggest this article to be published in the valuable journal of IJMS. However, as a minor comment, I suggest that the authors do a wider search and if there are other articles in this field, add their results to increase the number of references.

Response: Thanks for your comments. We added 40 references to this new manuscript.

Reviewer 3 Report

Comments and Suggestions for Authors

-The language in some portions are weak and therefore, a professional English language editor should check the MS to be checked carefully again.

-The abstract of the MS is not well documented to represent the whole manuscript objectives. The abstract should be arranged in the following order i.e. background of the research, aims, material and methods, key findings and conclusion. However, in the abstract section the aims and methods are not well clear please revised this part.

-Abbreviations used in whole manuscript have to be defined firstly and then their abbreviations have to be used.

- should be updated recent knowledge about disease (example cancer), nutraceutical applications and health-promoting of Chlorogenic acid in introduction part.

- include antioxidant of Chlorogenic acid

-Prepared and the specific mechanism of action (Chlorogenic acid) in one more Table  should be updated in the recent article

-Discussion of the MS is not appropriate and has to be modified using with some recent references. Author should include recent reference

-Please include one more section that is “Future trends and/or challenges” for the MS.

Comments on the Quality of English Language

The language in some portions are weak and therefore, a professional English language editor should check the MS to be checked carefully again.

Author Response

Response to Reviewer 3.

Comments and Suggestions for Authors

-The language in some portions are weak and therefore, a professional English language editor should check the MS to be checked carefully again.

Response: The manuscript was finally edited by a native English speaker.

-The abstract of the MS is not well documented to represent the whole manuscript objectives. The abstract should be arranged in the following order i.e. background of the research, aims, material and methods, key findings and conclusion. However, in the abstract section the aims and methods are not well clear please revised this part.

Response: this section was revised

-Abbreviations used in whole manuscript have to be defined firstly and then their abbreviations have to be used.

Response: The abbreviations were reviewed and a brief explanation was given the first time they were mentioned.

- should be updated recent knowledge about disease (example cancer), nutraceutical applications and health-promoting of Chlorogenic acid in introduction part.

Response: Information has been added in the introduction section.

- include antioxidant of Chlorogenic acid

Response: More information on the antioxidant activity of CGA was included in the chemoprotection section.

-Prepared and the specific mechanism of action (Chlorogenic acid) in one more Table  should be updated in the recent article

Response: The mechanisms of action of CGA for chemoresistance are listed in Table 2, and for chemoprotection are listed in the figure 4.

-Discussion of the MS is not appropriate and has to be modified using with some recent references. Author should include recent reference.

 Response: An additional 34 references were included in this new manuscript.

 -Please include one more section that is “Future trends and/or challenges” for the MS.

Response: New information about that was included in the conclusion

Reviewer 4 Report

Comments and Suggestions for Authors

Dear Authors,

The manuscript proposed may be of interest for the readers of International Journal of Molecular Sciences and represent a good contribution in the field of finding new natural adjuvant for cancer treatment.

The role of chlorogenic acid in reversal of chemoresistance and chemoprotection in cancer treatment is very well emphasized in this study.

The manuscript is well structured, concise, well documented, the references being in majority of current.

Please find below some suggestions:

1. Lines 90 and 95: I think it should be completed with beverages

- Ex: Functional foods and beverages high in CA

Table 1: CA composition in some vegetables and beverages.

Taking into account that carrots are also rich in CA, it could be mentioned (along with coffee and mate tea) their health benefits.

2. Conclusions: I suggest the elaboration in more detail of future trends indicating potential applications of chlorogenic acid.

Author Response

Response to Reviewer 4.

Dear Reviewer, We thank you for the comments and suggestions that you made. We improved this new manuscript by adding more than 40 references and including a new chapter.

Sincerely,

Dr. Cristian Paz

Comments and Suggestions for Authors

Please find below some suggestions:

  1. Lines 90 and 95: I think it should be completed with beverages

- Ex: Functional foods and beverages high in CA

Table 1: CA composition in some vegetables and beverages.

Taking into account that carrots are also rich in CA, it could be mentioned (along with coffee and mate tea) their health benefits.

Response: I apologize but we did not find enough information about beverages but we restructured this part, including valuable information about carrots and food processing, that increase the MS.

  1. Conclusions: I suggest the elaboration in more detail of future trends indicating potential applications of chlorogenic acid.

Response: New information about that was included in the conclusion

Round 2

Reviewer 3 Report

Comments and Suggestions for Authors

Accept in present form